# Electrospun Polyvinylpyrrolidone-Based Dressings Containing GO/ZnO Nanocomposites: A Novel Frontier in Antibacterial Wound Care

**DOI:** 10.3390/pharmaceutics16030305

**Published:** 2024-02-22

**Authors:** Cristina Martín, Adalyz Ferreiro Fernández, Julia C. Salazar Romero, Juan P. Fernández-Blázquez, Jabier Mendizabal, Koldo Artola, José L. Jorcano, M. Eugenia Rabanal

**Affiliations:** 1Department of Bioengineering, Universidad Carlos III de Madrid, 28911 Leganés, Spain; 100356873@alumnos.uc3m.es (J.C.S.R.); jjorcano@ing.uc3m.es (J.L.J.); 2Department of Materials Science and Engineering and Chemical Engineering & IAAB, Universidad Carlos III de Madrid, 28911 Leganés, Spain; adferrei@ing.uc3m.es; 3IMDEA Materials Institute, 28906 Getafe, Spain; juanpedro.fernandez@imdea.org; 4Domotek Ingeniería Prototipado y Formación S.L., 20003 San Sebastián, Spain; jabier@domotek.es (J.M.); info@domotek.es (K.A.); 5Instituto de Investigación Sanitaria Gregorio Marañón, 28007 Madrid, Spain

**Keywords:** electrospinning, graphene oxide, ZnO, antibacterial, dressings

## Abstract

In recent years, the rapid emergence of antibiotic-resistant bacteria has become a significant concern in the healthcare field, and although bactericidal dressings loaded with various classes of antibiotics have been used in clinics, in addition to other anti-infective strategies, this alarming issue necessitates the development of innovative strategies to combat bacterial infections and promote wound healing. Electrospinning technology has gained significant attention as a versatile method for fabricating advanced wound dressings with enhanced functionalities. This work is based on the generation of polyvinylpyrrolidone (PVP)-based dressings through electrospinning, using a DomoBIO4A bioprinter, and incorporating graphene oxide (GO)/zinc oxide (ZnO) nanocomposites as a potent antibacterial agent. GO and ZnO nanoparticles offer unique properties, including broad-spectrum antibacterial activity for improved wound healing capabilities. The synthesis process was performed in an inexpensive one-pot reaction, and the nanocomposites were thoroughly characterized using XRD, TEM, EDX, SEM, EDS, and TGA. The antibacterial activity of the dispersions was demonstrated against *E. coli* and *B. subtilis*, Gram-negative and Gram-positive bacteria, respectively, using the well diffusion method and the spread plate method. Bactericidal mats were synthesized in a rapid and cost-effective manner, and the fiber-based structure of the electrospun dressings was studied by SEM. Evaluations of their antibacterial efficacy against *E. coli* and *B. subtilis* were explored by the disk-diffusion method, revealing an outstanding antibacterial capacity, especially against the Gram-positive strain. Overall, the findings of this research contribute to the development of next-generation wound dressings that effectively combat bacterial infections and pave the way for advanced therapeutic interventions in the field of wound care.

## 1. Introduction

In recent years, wound care has undergone a transformative revolution driven by materials science and nanotechnology innovations [1,2]. Chronic wounds, surgical incisions, and traumatic injuries have posed persistent challenges to healthcare providers. Direct topical applications of antiseptics, silver- and iodine-based dressings, or medical-grade (Manuka) honey have been clinically tested as anti-infective strategies [3]. Regarding the use of antibiotics in wound dressings to reduce and eliminate the bioburden of local lesions, several classes of antibiotics, such as aminoglycosides, beta-lactams, glycopeptides, quinolones, sulfonamides, and tetracyclines, have been incorporated into mats and used in clinics [4]. However, in addition to the aforementioned complexities, there is an alarming increase in antibiotic-resistant bacterial infections, which require novel strategies for their effective treatment and prevention. Amid this backdrop, the intersection of electrospinning technology, nanomaterials, and polymers represents the possibility of pioneering developments based on composite-based bactericidal dressings, offering a unique and promising approach to tackle the pressing challenges associated with wound management.

Electrospinning, a versatile and powerful technique, stands at the forefront of this innovation [5]. It involves the controlled deposition of polymer fibers from a solution by applying an electric field, resulting in a fibrous scaffold characterized by an exceptionally high surface area-to-volume ratio [6]. This unique structural feature has opened up exciting possibilities for developing advanced wound dressings. In fact, the nanofibrous architecture closely mimics the extracellular matrix of human tissues, providing an ideal environment for cell adhesion, proliferation, and migration, thereby facilitating the wound healing process [7,8]. Additionally, the high porosity of electrospun mats allows for efficient moisture management, a crucial factor in promoting wound healing, and the flexibility of these dressings ensures ease of application and excellent conformity to the contours of wounds, enhancing patient comfort and overall efficacy. Finally, nanofiber scaffolds have emerged as a revolutionary drug delivery platform for promoting wound healing [7,9]. However, the use of nanofibers to achieve controlled drug loading and release still presents many challenges, with ongoing research still exploring how to load drugs onto nanofiber scaffolds without a loss of activity and how to control their release in a specific spatiotemporal manner. In this context, a few studies have already been published [10,11], and water-soluble electrospun nanofibers emerge as an optimal method. Polyvinylpyrrolidone (PVP), a common hydrophilic polymer [12] that is water-soluble, absorbs up to 40% of its weight under ambient conditions [13], and has good film-forming properties, is popular for the generation of electrospun mats. In this regard, Dai et al., for instance, reported the ability to store enzymes and other reagents on-chip in a rapidly dispersible format using PVP [14].

One of the most used strategies for the transformative power of electrospinning in wound care is the use of silver nanoparticles [15]. The study by Yang et al. [16], among others, demonstrated that these dressings exhibited excellent antibacterial properties against both Gram-positive and Gram-negative bacteria, making them promising candidates for infection control in wound care. However, despite their strong antibacterial activity, electrospinning dressings containing silver nanoparticles tend to be uneconomical to scale up. In this context, the integration of other bactericidal agents based on more affordable nanomaterials is essential. In this regard, graphene oxide (GO) and zinc oxide (ZnO) (two key nanomaterials offering antimicrobial attributes) are proposed in this work as candidates for the fabrication of PVP-based electrospun dressings as a game-changing strategy.

GO is a well-known two-dimensional graphene-based nanomaterial (GBN) with large surface area and excellent biocompatibility, generally due to its surface being decorated with oxygen-containing groups [17,18]. At the same time, various hypotheses on the bactericidal impact of GBNs, in general (i.e., the nanoblade effect, envelope, or oxidative stress induction), have been presented in the literature [19,20,21]. On the other hand, ZnO nanoparticles, reported by several studies as non-toxic to human cells [22], also exhibit antibacterial activity through different mechanisms such as the destabilization of microbial membranes upon the direct contact of ZnO particles to the cell walls [23], the generation of reactive oxygen species [24,25], or the release of zinc ions that disrupt bacterial membrane integrity and interfere with cellular processes [26]. When combined, GBNs and ZnO create synergistic effects, amplifying their individual antimicrobial properties [27,28].

Overall, this article explores PVP-based dressings containing GO/ZnO nanocomposites made by electrospinning, an advancement that could revolutionize the field of antibacterial wound treatment, especially in the context of antibiotic-resistant infections.

## 2. Experimental Methods

### 2.1. Materials

Zinc acetate dihydrate Zn(Ac)_2_·2H_2_O and sodium hydroxide were purchased from Sigma Aldrich, (St. Louis, MO, USA), and used as received without further purification. The aqueous dispersion of graphene oxide (GO) was provided by our collaborators (Asturias, Spain). PVP powder (Average Molecular Wt. 360,000) was purchased from TCI EUROPE N.V., Belgium. Luria Broth Base (Miller’s LB Broth Base)™ and LB agar, were purchased from Sigma Aldrich, USA, and Fisher Scientific, respectively. *Escherichia coli* (lyophilized cells) and *Bacillus subtilis* 1904-E were purchased from Merck and ATCC, respectively.

### 2.2. Synthesis of GO/ZnO Nanocomposites

A simple one-pot synthesis method was used to achieve the GO/ZnO nanocomposite materials based on GO and Zn(Ac)_2_·2H_2_O with 1:1 and 2:1 *w*/*w* ratios (GO/ZnO_1:1 and GO/ZnO_2:1, respectively). Briefly, 10 mL of the GO suspension was taken, and the corresponding amount of the Zn precursor (Zn(Ac)_2_·2H_2_O) was added to obtain the corresponding ratios in our study. In addition, a 0.6 M NaOH suspension was added to maintain the basic medium and to favor the formation of GO/ZnO nanocomposites. The reaction was kept for three hours under vigorous stirring conditions. The resulting nanocomposites were washed in distilled water several times until reaching neutral a pH, then washed with pure ethanol. GO/ZnO nanocomposites were used as freeze-dried powders.

### 2.3. Characterization of the Nanomaterials 

X-ray diffraction (XRD) analysis was performed to identify the crystalline phases presented by means of the diffraction patterns of the samples and nanocomposites by a Philips 30XL (SFEG, Nederland) with CuK α radiation, λ = 1.5418 Å, under a voltage of 40 kV and a current of 40 mA. The diffraction data of samples were recorded for 2θ angles between 7 and 55. The transmission electron microscope used to study the surface morphology of the nanomaterials and to perform the Energy Dispersive X-Ray (EDX) analyses to confirm the presence of Zn was a JEOL JEM 2010 coupled to an XEDS microanalysis system (Oxford Inca, France), with an accelerating voltage of 200 kV and a resolution between points of 0.25 nm. EDS (Energy Dispersive Spectrometer) analysis to determine the elemental compositions of the nanomaterials was performed using a Philips XL-30 conventional Scanning Electron Microscope coupled with an SDD-type EDS detector for microanalysis. TGA curves of the freeze-dried dispersions of the tested nanomaterials to study the thermal stability of the samples were acquired by using a TGA Q50 instrument (TA Instruments Company, New Castle, DE, USA) from 30 to 900 °C with a ramp of 10 °C/min under N_2_ or air using a flow rate of 90 mL/min and platinum pans. 

### 2.4. Synthesis of the Electrospun Dressings 

PVP (13% *w*/*v*) was dissolved in pure ethanol. Then, to prepare the inks containing the different carbon-based nanomaterials, the latter were dispersed in the previously prepared PVP solution at a final concentration of 5 mg/mL. Both GO/ZnO_1:1 and GO/ZnO_2:1 nanocomposites were used as freeze-dried powders. The aqueous dispersion of GO was used directly for the GO control dressing. A second control sample consisted of a dressing based only on PVP (13% *w*/*v*). Once the different solutions (inks) were prepared, a DomoBio 4A bioprinter (Domotek S.L., Gipuzkoa, Spain) at room temperature (18–22 °C) was used to create the electrospun mats. The experimental setup involved applying a voltage of 10 kV, maintaining a flow rate of 10 mL/min, using a needle with a diameter of 0.4 mm, and positioning the needle 80 mm away from the collector. After nine minutes of spinning, the dressing samples were cut into round shapes (discs) with a 0.8 cm diameter punch for further characterization and testing.

### 2.5. SEM Characterization of the Electrospun Discs 

The fibers’ surface was analyzed by SEM (Philips XL-30 conventional Scanning Electron Microscope) operating at 10 kV. Fiber diameters were calculated by the ImageJ program.

### 2.6. Antibacterial Studies

The well diffusion method first proved the antibacterial ability of all the nanomaterials. GO (10.0 mg/mL), GO/ZnO_1:1 (0.1 mg/mL), GO/ZnO_2:1 (0.1 mg/mL), and ZnO control (1.2 mg/mL) dispersions were dropped in 8 mm diameter wells made on previously seeded agar plates using *Escherichia coli (E. coli*) and *Bacillus subtilis* (*B. subtilis*). In addition, a positive control of H_2_O_2_ solution (1.0 M) poured onto a round filter paper was added to ensure that the experiment was carried out correctly. The spread plate method was also performed to confirm the bactericidal capacity of our dispersions. Firstly, both bacteria strains were grown in Lysogeny Broth (LB) medium at 37 °C under 210 rpm shaking speed and turbidity was adjusted to 1.9 × 10^5^ CFU/mL (O.D. was measured with a Biowave II spectrophotometer (Biochrom, Cambridge, UK at 600 nm). The cells were harvested by centrifugation, washed twice with PBS, and resuspended in the appropriate saline medium. Both strains were incubated with different freshly prepared nanomaterial dispersions (final concentration of 0.1 mg/mL) in PBS at 37 °C under a shaking speed of 210 rpm for two hours. Aliquots of samples were withdrawn, diluted, and then spread onto LB agar plates. After incubation at 37 °C, the capacity of the bacteria to form colonies was measured by quantifying the area of the bacteria (colonies) using the ImageJ program. All the treatments were performed at least in triplicate. 

Secondly, the disk-diffusion method was employed to corroborate the antibacterial abilities of the electrospun dressings containing (or not) the GO/ZnO nanocomposites. The dressing samples (8 mm in diameter) were irradiated with UV for 20 min to disinfect them. The films were then placed on the LB agar plates, which were inoculated with *E. coli* and *B. subtilis*, and the agar plates were incubated at 37 °C overnight. A positive control of H_2_O_2_ solution (1.0 M) poured onto a round filter paper was added to ensure that the experiment was performed correctly. The inhibition areas for each plate sample were photographed and quantitatively analyzed using the ImageJ 1.51s image software. Since the plate radius was known as 90 mm, the average plate radius in pixels was used to calculate the picture scale (pixel-to-mm ratio) [29]. 

## 3. Results and Discussion

### 3.1. Synthesis and Characterization of GO/ZnO Nanocomposites

A simple one-pot synthesis method was used to achieve the GO/ZnO nanocomposites based on GO and Zn(Ac)_2_·2H_2_O with 1:1 and 2:1 *w*/*w* ratios (GO/ZnO_1:1 and GO/ZnO_2:1, respectively). The composite nanostructures were fully characterized, and their antimicrobial activities were investigated. 

#### 3.1.1. Physicochemical Properties of GO/ZnO Nanocomposites

TEM micrographs of GO/ZnO_1:1 and GO/ZnO_2:1 nanocomposites are shown in Figure 1a,b and Figure 1c,d, respectively. As can be observed, ZnO nanoparticles form a star-shaped structure on the GO flakes. This specific morphology of the nanoparticles has been previously reported in the literature for ZnO composites at basic pH levels [30]. Interestingly, the star-shaped structure seems to be more diffused in the GO/ZnO_2:1 nanocomposite compared to in GO/ZnO_1:1, and it is also smaller in size. The higher GO/Zn(Ac)_2_·2H_2_O ratio could be influencing the formation of nanoparticles. Finally, and most importantly, the ZnO nanoparticles were well distributed throughout the entire GO sheets in both GO/ZnO_1:1 and GO/ZnO_2:1 nanocomposites (Appendix A). This uniform distribution is crucial for achieving desired properties and functionalities in nanocomposite materials. Furthermore, EDX experiments confirmed the presence of Zn when analyzing the ZnO nanoparticles (Appendix A). 

The elemental compositions (mass%) for the nanomaterials were measured by EDS analysis (Appendix A and Table 1). The results show that the elemental mass percent of Zn increased proportionally with its *w*/*w* proportion for the GO/ZnO nanocomposites. In addition, this analysis proved that both nanocomposites were successfully synthesized due to the presence of Zn, O, and C [31]. The carbon composition of GO/ZnO_1:1 was slightly lower due to the higher distribution of ZnO on the surface of GO, which was also shown in the microscopy analyses using TEM (Appendix A).

Figure 2a shows the TGA curves of GO and ZnO nanoparticles as control samples, as well as those of the GO/ZnO_1:1 and GO/ZnO_2:1 nanocomposites. In the case of GO, the first decomposition (100–150 °C) corresponded to the removal of physically adsorbed water molecules and the breakdown and loss of labile oxygen functionality groups. The second significant weight loss from 150 to 300 °C was due to the further decomposition of the oxygen-containing functional groups and to the carbon combustion [31,32]. The TGA profile of ZnO nanoparticles reveals a weight loss of 8% at about 150–260 °C, attributed to removing moisture content. No decomposition occurred after this up to 600 °C, which is in agreement with the literature [33]. The TGA curves obtained for both GO/ZnO_1:1 and GO/ZnO_2:1 achieved stability as well, due to the presence of the ZnO nanoparticles on the GO lattice surface, and, in agreement with EDS results, the nanocomposite composed of a higher content of Zn (GO/ZnO_1:1) displayed a lower weight loss at 300 °C (~5%) compared to that of GO/ZnO_2:1. As expected [34], this difference became more significant (~13%) when TGA was performed under an air atmosphere (Appendix A). 

The X-ray diffraction patterns from all the samples can be seen in Figure 2b. In the case of the GO control sample, it shows a diffraction maximum close to 2θ ≈ 10°, 9.88° (d-spacing: 8.8688 Å), corresponding to the (002) plane [35,36,37], indicating the presence of oxygen-containing groups. In the case of the ZnO control sample, the characteristic diffraction maxima corresponding to the wurtzite-type structure (JCPDS 891397) [38] are observed [39] at 31.73, 34.42, and 36.46, which matched well with the (100), (002), and (101) with hexagonal symmetry. The diffraction maxima of the graphene oxide and zinc oxide phases are noticed in the GO/ZnO composites. A variation in the relative intensities of diffraction maxima is observed due to the appearance of both phases, with a decrease in the intensities of the ZnO diffraction peaks as the proportion of GO in the nanocomposite is increased. The absence of any other peak confirms the purity of the nanocomposites. On the other hand, the absence of a diffraction peak (around 2θ ≈ 26°), corresponding to plane (002) of reduced graphene oxide (rGO) [40,41], indicated that a reduction in GO does not occur during the synthesis process, at least considerably or significantly enough to be detected by this technique.

#### 3.1.2. Antibacterial Studies of GO/ZnO-Based Nanocomposite Dispersions

The antibacterial activity of GO, ZnO nanoparticles, and GO/ZnO composites was first investigated qualitatively by the well diffusion technique against Gram-negative (i.e., *E. coli*) and Gram-positive (i.e., *B. subtilis*) bacteria (Appendix A). All dispersions showed antibacterial capacity against both strains, but a more significant effect was evident against *B. subtilis*, as more extensive inhibition areas were observed on plates grown with this strain compared to those with *E. coli*. The antibacterial studies were also performed using the plate test, as previously reported in the literature [20,42]. The antibacterial capacity was measured by quantifying the area of bacterial growth (Appendix A) using the ImageJ program, and the results obtained for both strains are shown in Figure 3. The incubation of both strains in the presence of the nanomaterials’ dispersions (0.1 mg/mL) led, in all cases, to a decrease in bacterial viability compared to the control samples incubated in the absence of any nanomaterial. Furthermore, the synergistic bactericidal effect achieved by the combination of GO and ZnO was evident for both GO/ZnO_1:1 and GO/ZnO_2:1, compared to the GO control alone, and, in agreement with the results from the well plate technique, *B. subtilis* resulted in a more susceptible strain to our nanocomposites than *E. coli*. 

### 3.2. Synthesis, Characterization, and Antibacterial Properties of GO/ZnO-Based Dressings

The electrospinning technique using a DomoBIO4A bioprinter (Domotek S.L., Gipuzkoa, Spain) to fabricate the GO/ZnO-based dressings in ambient conditions (18–22 °C) and a current of 10 kV for nine minutes to collect the fibers of the different inks (Figure 4). For further characterization and testing, the dressing samples were cut into round shapes with an 8 mm diameter punch.

All dressing samples were characterized by SEM, and fiber diameters were measured. Figure 5 shows representative images of the samples and the corresponding fiber diameter distributions.

The successful electrospinning of nanofibers is highly dependent on environmental factors such as relative humidity, and previous studies have emphasized the importance of humidity in controlling fiber diameter during electrospinning [43,44,45]. The possible variation in humidity levels during the electrospinning process could influence the diameter of the fibers, affecting the dressing’s overall structure, but also the specific nanomaterial itself [46]. As noted above, PVP absorbs up to 40% of its weight under ambient conditions [13], but once the polymer interacts with the nanomaterials, not only the relative humidity but also the different interactions between the polymer and the specific functional groups on the nanomaterial surface could alter the water absorption properties and ultimately interfere with the electrospinning process. In fact, despite the same electrospinning parameters being used in all the cases, the fiber diameter distribution of the GO/ZnO_2:1@PVP sample shifted to higher lengths compared to the PVP control and GO/ZnO_1:1@PVP samples. It seems that the higher the GO/ZnO ratio, the greater the thickness of the fibers and, consequently, the smaller the pore size between them. Moreover, in the case of the GO control sample, no fibers were even visible, and the dressings obtained for the GO control were hardly manipulable, heterogeneous, and almost transparent (Appendix A). In addition, not only the diameter but also the morphology of the fibers can be affected by the same factors discussed above. Actually, several factors (i.e., the concentration of the dispersion, applied electrical potential, flow rate, needle diameter, and needle-to-collector distance) affect not only the diameter, but also the distribution and alignment of the fibers [47,48]. Despite employing identical electrospinning conditions for all samples (see the Experimental Methods section for more details), electrical and surface forces between nanoparticles and the electric field generated during the process may vary depending on the nanoparticle content. This, in turn, affects the moisture levels and interactions between the polymer and nanomaterial, impacting solvent evaporation and the final mat morphology. In fact, In GO/ZnO_1:1, the fibers appear straight, while in GO/ZnO_2:1, the fibers exhibit a curled appearance (Figure 5).

The antimicrobial properties of the electrospun dressings were tested by the disk-diffusion method. All the films (8 mm in diameter) were placed on the LB agar plates previously inoculated with *E. coli* and *B. subtilis*. As expected, due to the high hydrophilicity of PVP, all dressings disappeared (dissolved) rapidly after contact with the wet agar gel, which is a great advantage from the point of view of the possible future clinical application of these dressings for disinfection and wound treatment. After the incubation of the agar plates containing the electrospun dressing samples at 37 °C, the inhibition areas for each sample on the plate were photographed (Appendix A) and quantitatively analyzed. Figure 6 displays the inhibition areas for each strain and each sample. 

The possible bactericidal capacity of PVP, alone or in combination with other polymers, has been previously demonstrated in the literature [49,50,51,52,53] and, in fact, a zone of inhibition appeared on the agar plate where the PVP control dressings was placed, resulting in 2707.4 ± 767.7 mm^2^ in the case of *E.coli* and double that (5566.4 ± 1057.9 mm^2^) in the case of *B. subtilis*. However, when combined with GO/ZnO nanocomposites, a significant enhancement was observed in the antibacterial activity: 3750.9 ± 824.3 mm^2^ and 5385.2 ± 1965.7 mm^2^ for *E. coli* and *B. subtilis*, respectively, in the case of GO/ZnO_1:1@PVP and 4633.1 ± 566.3 mm^2^ and 9861.9 ± 1244.8 mm^2^ for *E. coli* and *B. subtilis*, respectively, in the case of GO/ZnO_2:1@PVP. This enhancement could be attributed to several factors. First, the presence of ZnO nanoparticles in the nanocomposites may contribute to the intrinsic bactericidal capabilities of the PVP control and GO@PVP control dressings [54]. Second, as was previously explained, the interaction between the different nanomaterials, PVP, and ambient water molecules could play a crucial role in determining the fiber diameter during electrospinning, leading to variations in the dressing’s structure and density, and maybe in determining the rate of release of the antibacterial species. In fact, the obtention of the electrospun dressing GO@PVP control proved to be challenging. As was previously discussed, the resulting film was nearly transparent and devoid of fibers (see Appendix A). Despite using identical electrospinning conditions, the substantial difference in the dosage may also explain the lower antibacterial efficacy of GO-based dressings compared to those of the PVP control. 

Overall, it is essential to highlight that, under the conditions optimized in this work, GO/ZnO_1:1 and GO/ZnO_2:1 nanocomposites combined with PVP allowed us to obtain more convenient dressings using the electrospinning technique than when combined only with GO, and also to increase the bactericidal capacity of the virgin PVP dressing. Moreover, in line with the results obtained from the antimicrobial studies performed with the GO/ZnO nanocomposite-based dispersions (see Section 3.1.2), *B. subtilis* is more susceptible than *E. coli* when these strains are exposed to the dressings. 

Importantly, our nanocomposite system exhibits remarkable capabilities and potential synergistic effects, not limited to antibacterial activity alone. Other reported benefits include, but are not limited to, the following: some authors emphasize the significance of zinc oxide nanoparticles due to their ability to accelerate bone growth and mineralization. Additionally, they possess low toxicity, biocompatibility, bioactivity, and chemical stability. These biological properties render them potentially useful in orthopedic applications, demonstrating both antibacterial and osteogenic capacity. Therefore, investigating synergies with other co-decorated materials to enhance biological activity in implants is of considerable interest [55,56].

## 4. Conclusions

In conclusion, this study introduces a promising and groundbreaking approach to antibacterial wound care by utilizing novel electrospun PVP-based dressings containing GO/ZnO nanocomposites. These nanocomposites were synthesized in an easy, inexpensive, and single-step process and were thoroughly characterized, confirming the effectiveness of GO surface functionalization with ZnO nanoparticles, as well as the antibacterial properties of the dispersions against Gram-negative and Gram-positive strains, although outstandingly effective against *B. subtilis*. 

Under the conditions optimized in this study, we have demonstrated that the integration of GO/ZnO_1:1 and GO/ZnO_2:1 nanocomposites with PVP results in the fabrication of electrospun dressings rapidly and cost-effectively, surpassing the convenience and antibacterial efficacy of dressings composed solely of GO. The antibacterial ability of GO/ZnO nanocomposite-based dressings was demonstrated against *E. coli* and *B. subtilis*, again being most effective against the Gram-positive strain. This capacity could be attributed to the presence of ZnO nanoparticles and the influence of ambient humidity regarding the interactions between the nanocomposites and PVP on the fiber diameter during electrospinning. 

It is worth highlighting that this formulation stands out for its novelty, serving as a starting point for further analyses. Actually, this study paves the way for the development of advanced and cost-effective wound dressings with improved antibacterial properties for medical applications, presenting a promising avenue in the field of wound care that could revolutionize the treatment of infected wounds. 

## Figures and Tables

**Figure 1 pharmaceutics-16-00305-f001:**
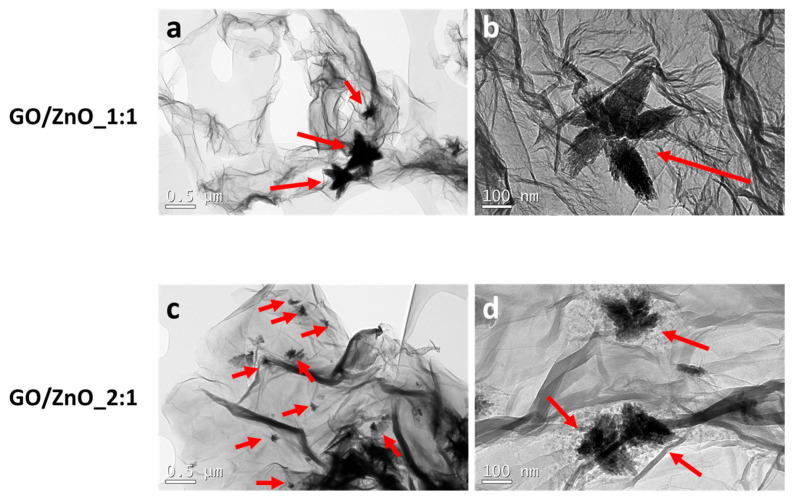
Representative TEM images of (**a**,**b**) GO/ZnO_1:1 and (**c**,**d**) GO/ZnO_2:1 nanocomposites. Scale bar (**a**,**c**): 0.5 µm. Scale bar (**b**,**d**): 100 nm. Red arrows indicate ZnO nanoparticles.

**Figure 2 pharmaceutics-16-00305-f002:**
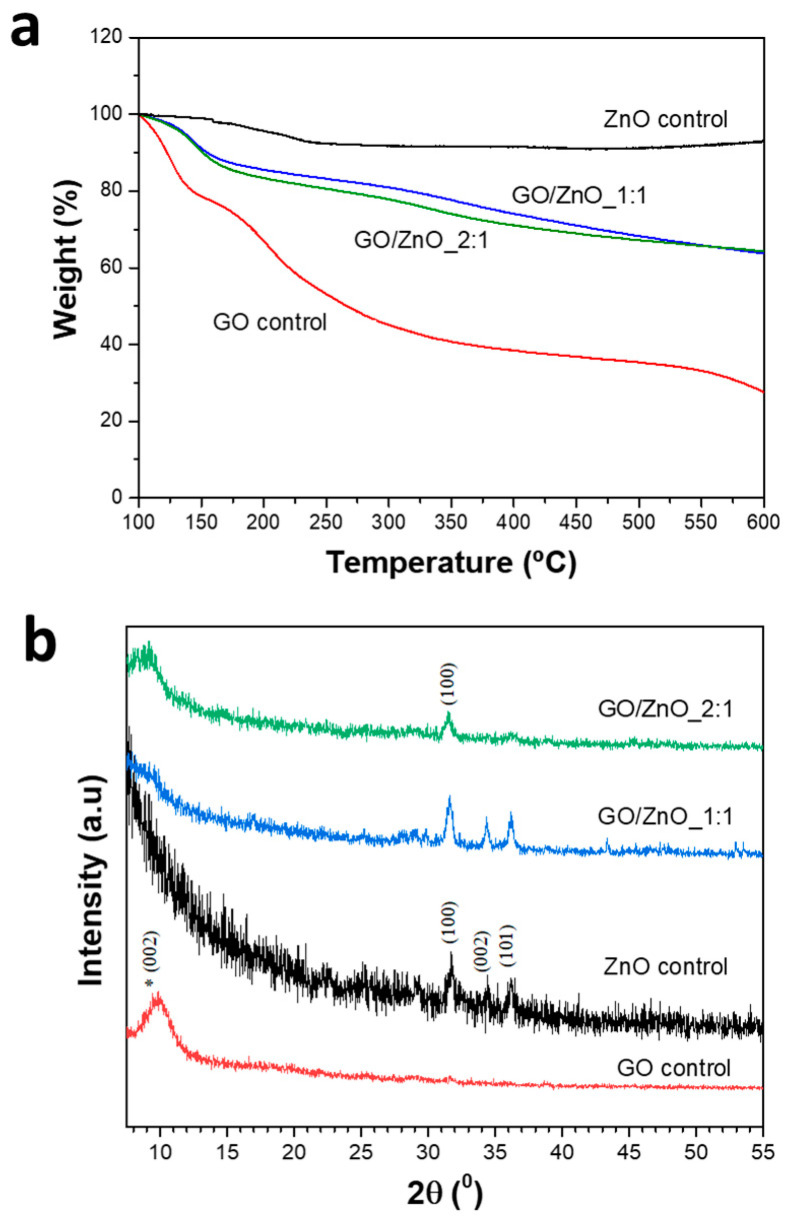
(**a**) TGA and (**b**) XRD analyses for GO, ZnO, GO/ZnO_1:1, and GO/ZnO_2:1 samples. The asterisk (*) indicates that the (002) plane is different between the ZnO control and GO control samples.

**Figure 3 pharmaceutics-16-00305-f003:**
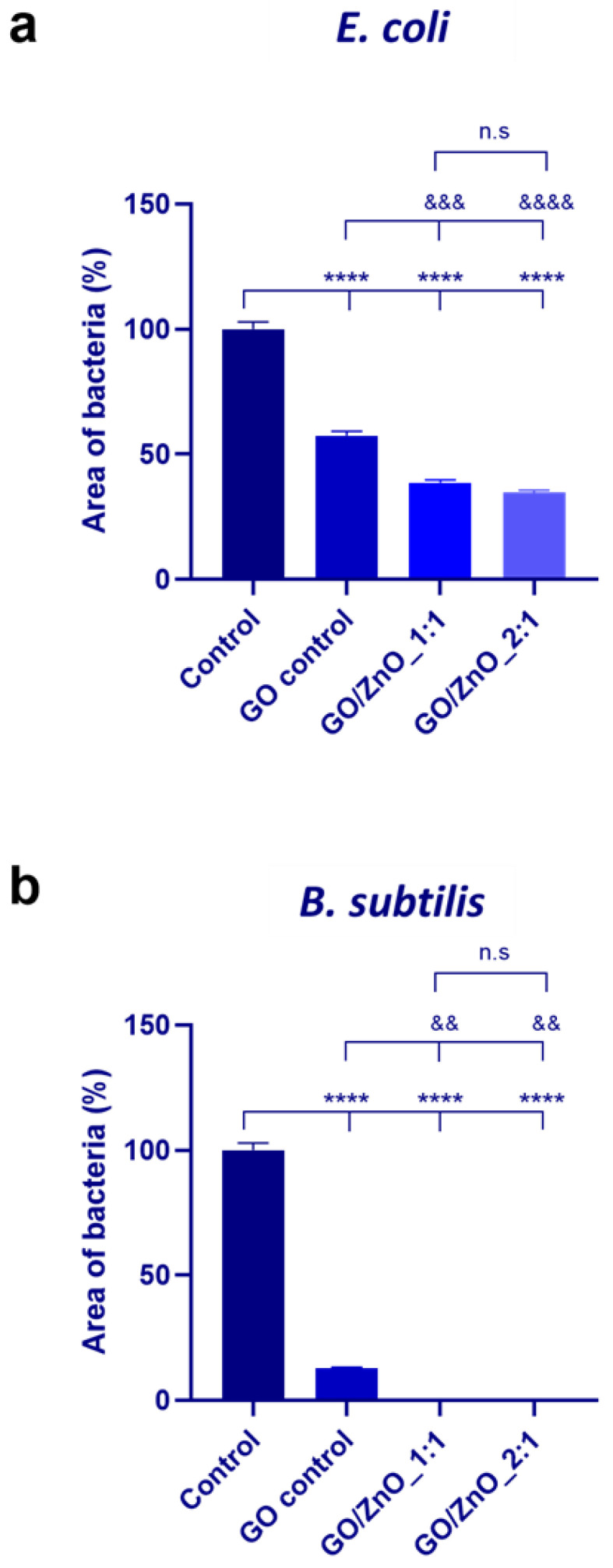
Bacterial viability quantified as the area of bacteria (%) of (**a**) *E. coli* and (**b**) *B. subtilis* grown on culture plates after incubation in the presence of the different nanomaterials’ dispersions (final concentration of 0.1 mg/mL). The control sample refers to the incubation of both strains in the absence of any nanomaterial. The results are expressed as average ± SEM for each material (*n* = 3). Statistical analysis was performed using One way Anova followed by Tukey’s multiple comparisons test; * denotes significant differences with respect to the control sample ((n.s. *p* > 0.05, * *p* < 0.05, ** *p* < 0.01, *** *p* < 0.001, **** *p* < 0.0001); & denotes significant differences with respect to the GO control sample (n.s. *p* > 0.05, & *p* < 0.05, && *p* < 0.01, &&& *p* < 0.001, &&&& *p* < 0.001)).

**Figure 4 pharmaceutics-16-00305-f004:**
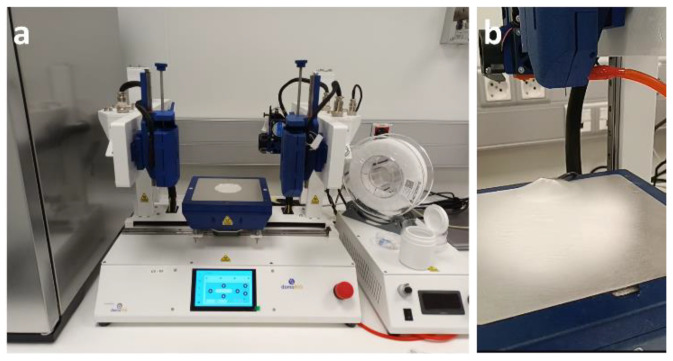
Digital pictures of (**a**) the DomoBio4A bioprinter and (**b**) a representative electrospun dressing of the PVP control sample.

**Figure 5 pharmaceutics-16-00305-f005:**
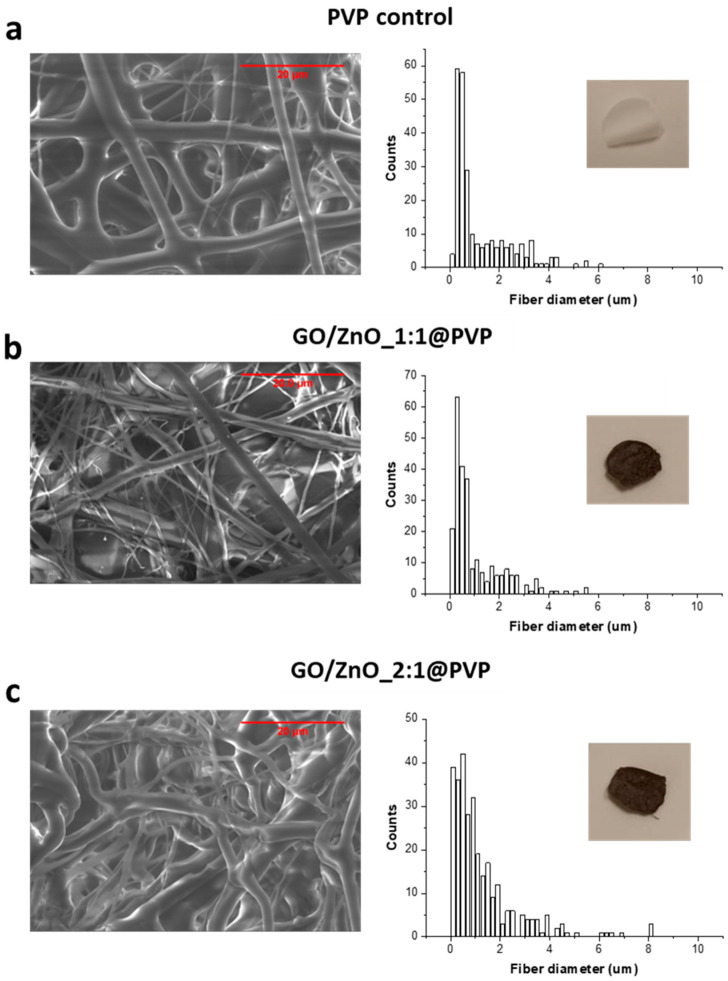
Representative SEM images (left panel) and fiber diameter distributions (right panel) of the electrospun dressings: (**a**) PVP control, (**b**) GO/ZnO_1:1@PVP, and (**c**) GO/ZnO_2:1@PVP. Scale bars: 20 µm. The inset images show digital photos of the dressing samples cut in round shapes with a diameter of 8 mm.

**Figure 6 pharmaceutics-16-00305-f006:**
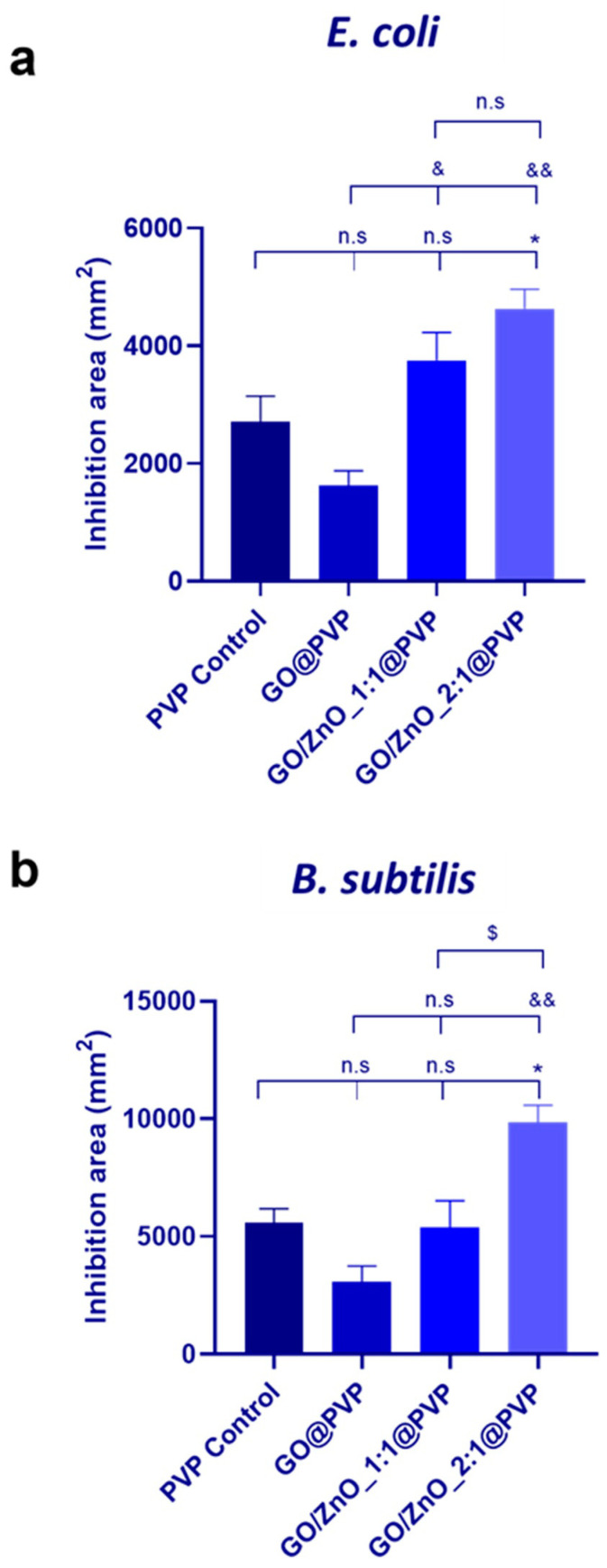
Antibacterial capacity quantified as inhibition area (mm^2^) of (**a**) *E. coli* and (**b**) *B. subtilis* grown on LB agar plates after incubation in the presence of different electrospun dressings. Results are expressed as average ± SEM for each material (*n* = 3). Statistical analysis was performed using One way Anova followed by Tukey’s multiple comparisons test; * denotes significant differences with respect to the PVP control ((n.s. *p* > 0.05, * *p* < 0.05, ** *p* < 0.01, *** *p* < 0.001, **** *p* < 0.0001), and & denotes significant differences with respect to the GO@PVP control sample (n.s. *p* > 0.05, & *p* < 0.05, && *p* < 0.01, &&& *p* < 0.001, &&&& *p* < 0.001)). $ denotes significant differences between ratios.

**Table 1 pharmaceutics-16-00305-t001:** EDS analyses for GO, ZnO, GO/ZnO_1:1, and GO/ZnO_2:1 samples.

Element (mass%)			
Sample	C	O	Zn
GO control	74.3	22.8	
ZnO control		24.7	75.3
GO/ZnO_1:1	56.2	31.8	9.4
GO/ZnO_2:1	55.9	29.2	5.9

## Data Availability

The raw data supporting the conclusions of this article will be made available by the authors on request.

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
