# Peer review of "Electrospun Polyvinylpyrrolidone-Based Dressings Containing GO/ZnO Nanocomposites: A Novel Frontier in Antibacterial Wound Care"

_pharmaceutics, 2024, doi:10.3390/pharmaceutics16030305_

Round 1

Reviewer 1 Report

Comments and Suggestions for Authors

The paper deals with the development of a novel PVP-based dressings containing nanocomposites. Although the composite dressings is interesting with considerable bactericidal applications, the quality of the paper needs to improve before accepting it by the journal. My specific comments are as follows:

1.       While drug resistance is prevalent in clinical settings, I do not believe that a significant amount of antibiotics is utilized in typical wound dressing designs. Consequently, there is a need for a more precise assessment of the true value of synthetic dressings in the context of abstract and introduction.

2.       Line 212: SEM and element mapping results of composites should be added to fig.2.

3.       Line 213: clerical error of XRD.

4.       Line 230 and 304: the antibacterial evaluation of dispersions and electro spun dressings are not reasonable. First, the results should include statistical analysis to illustrate the significance of the difference. Second, The antibacterial results should include time-dependent and concentration-dependent aspects. Last but not least, as different material form, X-axis label of fig.6 should be modified with GO@PVP and GO/ZnO(1:1)@PVP, etc.

5.       The discussion section is inadequate and needs to be rewritten.

6.       The quality of some charts need to be improved to meet the standard for publishing.

Comments on the Quality of English Language

Moderate editing of English language required

Author Response

Reviewer 1

The paper deals with the development of a novel PVP-based dressings containing nanocomposites. Although the composite dressings is interesting with considerable bactericidal applications, the quality of the paper needs to improve before accepting it by the journal. My specific comments are as follows:

Response: We thank the reviewer 1 for his/her comments, which helped us to improve the quality of the final manuscript. A point-by-point answer can be found below:

Comment 1.       While drug resistance is prevalent in clinical settings, I do not believe that a significant amount of antibiotics is utilized in typical wound dressing designs. Consequently, there is a need for a more precise assessment of the true value of synthetic dressings in the context of abstract and introduction.

Response: we acknowledge the importance of his/her concern regarding the precise assessment of synthetic dressings in the context of drug resistance. Although Reviewer 1 does not believe that a significant amount of antibiotics is utilized in typical wound dressings, several classes of antibiotics, including aminoglycosides, beta-lactams, glycopeptides, quinolones, sulfonamides, and tetracyclines, have been incorporated into mats and used in the clinic (https://doi.org/10.1016/j.ejpb.2018.02.022). To address this comment, we have revised the abstract and introduction sections to provide a more comprehensive overview of the current landscape of antibiotic usage in wound dressing designs.

“…healthcare field, and although bactericidal dressings loaded with various classes of antibiotics have been used in the clinic, in addition to other anti-infective strategies, this alarming issue…”

"…healthcare providers. Direct topical application of antiseptics, silver- and iodine-based dressings, or medical-grade (Manuka) honey have been clinically tested as anti-infective strategies.3 Regarding the use of antibiotics in wound dressings to reduce and eliminate the bioburden of local lesions, several classes of antibiotics, such as aminoglycosides, beta-lactams, glycopeptides, quinolones, sulfonamides and tetracyclines, have been incorporated into mats and used in the clinic.4 However, in addition to the aforementioned complexities, there is an alarming increase in antibiotic-resistant bacterial infections, which require novel strategies for their effective treatment and prevention. Amid this backdrop,…”

Moreover, it is worth highlighting that this formulation stands out for its novelty, serving as a starting point for further analysis. This paper presents a preliminary investigation of these mats, providing a solid foundation that can be used to achieve further results in the future. Importantly, our nanocomposite system exhibits remarkable capabilities and potential synergistic effects, not limited to antibacterial activity alone. Other reported benefits include, but are not limited to, the following: some authors emphasize the significance of zinc oxide nanoparticles due to their ability to accelerate bone growth and mineralization. Additionally, they possess low toxicity, biocompatibility, bioactivity, and chemical stability. These biological properties render them potentially useful in orthopedic applications, demonstrating both antibacterial and osteogenic capacity. Therefore, investigating synergies with other co-decorated materials to enhance biological activity in implants is of considerable interest. We have now introduced this information into the revised manuscript:

“Importantly, our nanocomposite system exhibits remarkable capabilities and potential synergistic effects, not limited to antibacterial activity alone. Other reported benefits include, but are not limited to, the following: some authors emphasize the significance of zinc oxide nanoparticles due to their ability to accelerate bone growth and mineralization. Additionally, they possess low toxicity, biocompatibility, bioactivity, and chemical stability. These biological properties render them potentially useful in orthopedic applications, demonstrating both antibacterial and osteogenic capacity. Therefore, investigating synergies with other co-decorated materials to enhance biological activity in implants is of considerable interest.55,56

“It is worth highlighting that this formulation stands out for its novelty, serving as a starting point for further analyses. Actually, this study paves the way…”

Comment 2.       Line 212: SEM and element mapping results of composites should be added to fig.2.

Response: SEM pictures and element mapping results of GO/ZnO_1:1 and GO/ZnO_2:1   composites have now been included as Figures S3 and S4, instead of being added to Figure 2, due to the large size of the figures. The numbering of the figures that needed to be modified has also been corrected.

Figure S3. SEM pictures and element mapping results of GO/ZnO_1:1 composite.

Figure S4. SEM pictures and element mapping results of GO/ZnO_2:1 composite.

Comment 3.       Line 213: clerical error of XRD.

Response: the clerical error of XRD has been corrected:

“… XRD analyses for GO, ZnO, GO/ZnO_1:1 and GO/ZnO_2:1 samples.”

Comment 4.       Line 230 and 304: the antibacterial evaluation of dispersions and electro spun dressings are not reasonable. First, the results should include statistical analysis to illustrate the significance of the difference. Second, The antibacterial results should include time-dependent and concentration-dependent aspects. Last but not least, as different material form, X-axis label of fig.6 should be modified with GO@PVP and GO/ZnO(1:1)@PVP, etc.

Response: we acknowledge reviewer 1 for this comment, as it will help us to improve Figures 3 and 6 and clarify some points. First, statistical analysis has been added to Figures 3 and 6 to illustrate the significance of the findings. Additionally, we have modified the nomenclature of the dressings in the entire manuscript as indicated. Regarding the antibacterial evaluation, we acknowledge his/her points. However, it's important to clarify that our study's primary focus was not on exploring time and dose-dependent responses beyond nanoparticle loading. Instead, these initial results serve as a foundational exploration. As previously indicated in comment 1, this paper presents a preliminary investigation of these mats, providing a solid foundation that can be used to achieve further results and we will certainly consider your suggestions for future investigations. As for the X-axis labels in Figure 6, we have revised them to accurately reflect the different material forms, including GO@PVP and GO/ZnO_1:1@PVP, etc… as he/she suggested.

Figure 3. Bacterial viability quantified as the area of bacteria (%) of (a) E. coli and (b) B. subtilis grown on culture plates after incubation in the presence of the different nanomaterials’ dispersions (final concentration of 0.1 mg/mL). The control sample refers to the incubation of both strains in the absence of any nanomaterial. The results are expressed as average ± SEM for each material (n = 3). Statistical analysis was performed using One way Anova followed by Tukey's multiple comparisons test, * denotes significant differences respect to the Control sample ((p > 0.05, * p < 0.05, ** p < 0.01, *** p < 0.001, **** p < 0.0001), & denotes significant difference respect to the GO control sample (p > 0.05, & p < 0.05, && p < 0.01, &&& p < 0.001, &&&& p < 0.001)).

Figure 6. Antibacterial capacity quantified as Inhibition area (mm2) of (a) E. coli and (b) B. subtilis grown on LB agar plates after incubation in the presence of the different electrospun dressings. The results are expressed as average ± SEM for each material (n = 3). Statistical analysis was performed using One way Anova followed by Tukey's multiple comparisons test, * denotes significant differences respect to the PVP control ((p > 0.05, * p < 0.05, ** p < 0.01, *** p < 0.001, **** p < 0.0001), & denotes significant difference respect to the GO@PVP control sample (p > 0.05, & p < 0.05, && p < 0.01, &&& p < 0.001, &&&& p < 0.001)). $ denotes significant differences between the ratios.

Comment 5.       The discussion section is inadequate and needs to be rewritten.

Response: we have carefully reviewed the discussion section and made several revisions to the text. These parts have been marked in yellow throughout the section. However, regarding the part of the text that you found "inadequate," we would greatly appreciate it if he/she could provide more specific details on what aspects he/she found lacking or problematic. Understanding his/her concerns in more detail would allow us to address them effectively and further refine the manuscript to even more closely meet the standards of rigor and precision expected in scientific research.

Comment 6.       The quality of some charts need to be improved to meet the standard for publishing.

Response: we have already reviewed several graphs and images, including Figures 1, 3, 5 and 6, to ensure that they meet publication standards (the new Figures 3 and 6 have already been shown in comment 4). However, once again, we would appreciate it if he/she could provide us with more specific details on the charts that, in his/her opinion, require further improvement. Any additional guidance he/she can provide will help us make the necessary adjustments to meet his/her expectations and ensure the clarity and effectiveness of our visuals.

Figure 1. Representative TEM images of (a,b) GO/ZnO_1:1 and (c,d) GO/ZnO_2:1 nanocomposites. Scale bar (a,c): 0.5 µm. Scale bar (b,d): 100 nm. Red arrows indicate ZnO nanoparticles.

Figure 5. Representative SEM images (left panel) and fiber diameter distributions (right panel) of the electrospun dressings: a) PVP control, b) GO/ZnO_1:1@PVP, and c) GO/ZnO_2:1@PVP. Scale bars: 20 µm. The inset images show digital photos of the dressing samples cut in round shapes with a diameter of 8 mm.

Reviewer 2 Report

Comments and Suggestions for Authors

In this article, the authors reported an electrospinning technology fabricating PVP based dressings decorated by graphene oxide (GO)-zinc oxide (ZnO) as a potent antibacterial agent. The authors use XRD, TEM, EDX, SEM, EDS, and TGA to figure out the properties of. the composite films. The antibacterial activities of GO and ZnO were demonstrated against E. coli and B. subtilis. The reported GO/ZnO nanocomposites exhibit promising potential for applications in the development of advanced wound healing materials.

Some questions and suggestions for improvement:

1.     Distinguishing between GO and ZnO in Figure 1 is challenging. To enhance clarity, I recommend that the authors include separate images for single ZnO and GO.

2.     The morphology of the nanofibers undergoes significant changes with varying GO/ZnO compositions. In GO/ZnO_1:1, the fibers appear straight, while in GO/ZnO_2:1, the fibers exhibit a curled appearance. The authors may consider addressing the reasons behind these morphological variations.

3.     In Figure 6, the inhibition areas of GO control for both E.coli and B.subtilis are less than PVP control , what’s the reason?

Comments on the Quality of English Language

English is good!

Author Response

Reviewer 2

In this article, the authors reported an electrospinning technology fabricating PVP based dressings decorated by graphene oxide (GO)-zinc oxide (ZnO) as a potent antibacterial agent. The authors use XRD, TEM, EDX, SEM, EDS, and TGA to figure out the properties of. the composite films. The antibacterial activities of GO and ZnO were demonstrated against E. coli and B. subtilis. The reported GO/ZnO nanocomposites exhibit promising potential for applications in the development of advanced wound healing materials.

Response: We appreciate his/her thoughtful review of our manuscript.

Some questions and suggestions for improvement:

Comment 1.     Distinguishing between GO and ZnO in Figure 1 is challenging. To enhance clarity, I recommend that the authors include separate images for single ZnO and GO.

Response: we thank reviewer 2 for this comment, as it helped us to improve the Figure 1 for better understanding.

Figure 1. Representative TEM images of (a,b) GO/ZnO_1:1 and (c,d) GO/ZnO_2:1 nanocomposites. Scale bar (a,c): 0.5 µm. Scale bar (b,d): 100 nm. Red arrows indicate ZnO nanoparticles.

Comment 2.     The morphology of the nanofibers undergoes significant changes with varying GO/ZnO compositions. In GO/ZnO_1:1, the fibers appear straight, while in GO/ZnO_2:1, the fibers exhibit a curled appearance. The authors may consider addressing the reasons behind these morphological variations.

Response: we thank reviewer 2 again for this comment, as it has helped to improve and clarify this point of the discussion.

Several factors (i.e. concentration of the dispersion, applied electrical potential, flow rate, needle diameter, and needle to collector distance) affect not only the diameter, but also the distribution and alignment of the fibers. All samples were electrospun under the same conditions. However, it is certain that the electrical and surface forces between the nanoparticles and the electric field generated during the electrospinning process could be modified depending on the amount of nanoparticles contained in the sample, affecting the evaporation of the solvent and, therefore, the final morphology of the mat (doi: 10.1007/s10965-013-0105-9)(doi: 10.1016/j.progpolymsci.2013.02.001).

We have now addressed the reasons in the main manuscript:

“…almost transparent (Figure S8). In addition, not only the diameter but also the morphology of the fibers can be affected by the same factors discussed above. Actually, several factors (i.e. concentration of the dispersion, applied electrical potential, flow rate, needle diameter, and needle to collector distance) affect not only the diameter, but also the distribution and alignment of the fibers.47,48 Despite employing identical electrospinning conditions for all samples (see methods section for more details), electrical and surface forces between nanoparticles and the electric field generated during the process may vary depending on the nanoparticle content. This, in turn, affects moisture levels and interactions between the polymer and nanomaterial, impacting solvent evaporation and the final mat morphology.  In fact, In GO/ZnO_1:1, the fibers appear straight, while in GO/ZnO_2:1, the fibers exhibit a curled appearance (Figure 5).”

“…electrospun mats. The experimental setup involved applying a voltage of 10 kV, maintaining a flow rate of 10 mL/min, using a needle with a diameter of 0.4 mm, and positioning the needle 80 mm away from the collector. After nine minutes…”

Comment 3.     In Figure 6, the inhibition areas of GO control for both E.coli and B.subtilis are less than PVP control , what’s the reason?

Response: as already discussed in the manuscript:

“Second, as it has been previously explained, the interaction between the different nanomaterials, PVP and ambient water molecules could be playing a crucial role in determining the fiber diameter during electrospinning, leading to variations in the dressing’s structure and density, and maybe in determining the rate of release of the antibacterial species. In fact, the obtention of the electrospun dressing GO@PVP control proved to be challenging. As it has been previously discussed, the resulting film was nearly transparent and devoid of fibers (see Figure S8). Despite using identical electrospinning conditions, the substantial difference in the dosage may also explain the lower antibacterial efficacy of GO-based dressings compared to those of PVP control.”

In a few words, our hypothesis is that the intrinsic bactericidal capacity of PVP observed in the PVP control mat is attenuated in the presence of GO possibly due to a substantial difference in the dosage because of the difficulties in obtaining the GO@PVP dressing.

Reviewer 3 Report

Comments and Suggestions for Authors

The manuscript "Electrospun PVP-Based Dressings Containing GO/ZnO Nanocomposites: A Novel Frontier in Antibacterial Wound Care" is interesting and suitable for Pharmaceutics. The authors need to make some changes and clarifications:

1. Abstract:

- Lines 25-26 are a repetition of lines 17-19. Please reformulate.

- Lines 25-29 would need to be rewritten, they should be presented as conclusions of the study.

2. It would be preferable for the experimental methods to be in subsections.

3. Lines 163-167, I think they are not essential.

4. The data from Figure 2a should be presented in a Table.

5. For Figures 5a, b and c (right panel), it is preferable to have the same scale.

6. Nanocomposites with PVP should be named differently. In section 3.2 it is very difficult to differentiate the samples with PVP, especially in Figures 5 and 6.                                                                                                                                                                                                                                                                                                                                                                                                                                                                                   

Author Response

Reviewer 3

The manuscript "Electrospun PVP-Based Dressings Containing GO/ZnO Nanocomposites: A Novel Frontier in Antibacterial Wound Care" is interesting and suitable for Pharmaceutics. The authors need to make some changes and clarifications:

Response: we wish to thank the referee for the positive comment to our work. A point-by-point answer can be found below:

Comment 1. Abstract:

- Lines 25-26 are a repetition of lines 17-19. Please reformulate.

- Lines 25-29 would need to be rewritten, they should be presented as conclusions of the study.

Response: the abstract has been reformulated as follows:

“The antibacterial activity of the dispersions was demonstrated against E. coli and B. subtilis, Gram-negative and Gram-positive bacteria, respectively, using the well diffusion method and the spread plate method. Bactericidal mats were synthesized in a rapid and cost-effectively manner, and the fiber-based structure of the electrospun dressings was studied by SEM. The evaluation of their antibacterial efficacy against E. coli and B. subtilis were explored by the disk-diffusion method, revealing an outstanding antibacterial capacity, especially against the Gram-positive strain.”

Comment 2. It would be preferable for the experimental methods to be in subsections.

Response: the experimental methods section is already divided in different subsections (i.e. Materials, Synthesis of GO/ZnO nanocomposites, Characterization of the nanomaterials, Synthesis of the electrospun dressings, SEM characterization of the electrospun discs, and Antibacterial studies). We have now changed the format and included the numbering to these sections.

“2.1. Materials”

“2.2. Synthesis of GO/ZnO nanocomposites”

“2.3. Characterization of the nanomaterials”

“2.4. Synthesis of the electrospun dressings”

“2.5. SEM characterization of the electrospun discs”

“2.6. Antibacterial studies”

Comment 3. Lines 163-167, I think they are not essential.

Response: we thank reviewer 3 for his comment. However, in our opinion, lines 163-167 serve as a brief introduction to the section "3.1. Synthesis and Characterization of GO/ZnO Nanocomposites" in which we emphasize one of the advantages of this article: the one-step synthesis of the nanocomposites, and we also recall the labeling of the nanocomposites in terms of nanoparticle content, useful information to understand the following sections of the manuscript. Therefore, although we agree with reviewer 3 that they are not essential, we think this short introduction will favor the comprehension of the subsequent sections of the article.

Comment 4. The data from Figure 2a should be presented in a Table.

Response: the data from Figure 2a have been presented in a Table 1:

Table 1. EDS analyses for GO, ZnO, GO/ZnO_1:1 and GO/ZnO_2:1 samples.

element (mass%)

sample

C

O

Zn

GO control

74.3

22.8

ZnO control

24.7

75.3

GO/ZnO_1:1

56.2

31.8

9.4

GO/ZnO_2:1

55.9

29.2

5.9

Comment 5. For Figures 5a, b and c (right panel), it is preferable to have the same scale.

Response: Figures 5a, b and c (right panel) have the same scale: 20 um. Moreover, according to comment 6, the labeling of the dressings has been modified:

Figure 5. Representative SEM images (left panel) and fiber diameter distributions (right panel) of the electrospun dressings: a) PVP control, b) GO/ZnO_1:1@PVP, and c) GO/ZnO_2:1@PVP. Scale bars: 20 µm. The inset images show digital photos of the dressing samples cut in round shapes with a diameter of 8 mm.

Comment 6. Nanocomposites with PVP should be named differently. In section 3.2 it is very difficult to differentiate the samples with PVP, especially in Figures 5 and 6.       

Response: we thank reviewer 3 for this comment, as it helped us to improve the figures and clarify this point. As previously mentioned in comment 5, we have modified the labels of the nanocomposites with PVP, both in the text and in the figures, by adding “@PVP” (i.e. GO@PVP, GO/ZnO_1:1@PVP and GO/ZnO_2:1@PVP). Figure 5 has been already shown in the previous comment, and Figures 3 and 6 look as follows:

Figure 3. Bacterial viability quantified as the area of bacteria (%) of (a) E. coli and (b) B. subtilis grown on culture plates after incubation in the presence of the different nanomaterials’ dispersions (final concentration of 0.1 mg/mL). The control sample refers to the incubation of both strains in the absence of any nanomaterial. The results are expressed as average ± SEM for each material (n = 3). Statistical analysis was performed using One way Anova followed by Tukey's multiple comparisons test, * denotes significant differences respect to the Control sample ((p > 0.05, * p < 0.05, ** p < 0.01, *** p < 0.001, **** p < 0.0001), & denotes significant difference respect to the GO control sample (p > 0.05, & p < 0.05, && p < 0.01, &&& p < 0.001, &&&& p < 0.001)).

Figure 6. Antibacterial capacity quantified as Inhibition area (mm2) of (a) E. coli and (b) B. subtilis grown on LB agar plates after incubation in the presence of the different electrospun dressings. The results are expressed as average ± SEM for each material (n = 3). Statistical analysis was performed using One way Anova followed by Tukey's multiple comparisons test, * denotes significant differences respect to the PVP control ((p > 0.05, * p < 0.05, ** p < 0.01, *** p < 0.001, **** p < 0.0001), & denotes significant difference respect to the GO@PVP control sample (p > 0.05, & p < 0.05, && p < 0.01, &&& p < 0.001, &&&& p < 0.001)). $ denotes significant differences between the ratios.

Reviewer 4 Report

Comments and Suggestions for Authors

The research manuscript reported by Martín et al. demonstrated that the PVP-based nanofibers co-loaded with GO and ZnO nanoparticles are potential wound dressing materials for the treatment of infected wounds. Due to the lot of data that is missing, the article is not yet suitable for publication in pharmaceutics because there is a lot of data that is missing. However, the authors must address the following comments and suggestion before the paper can be considered for further review processes:

1.      The underlined subtitles under experimental methods should be numbered from 2.1. to 2.5. instead of underlined.

2.      The reason for the application of each characterization technique (e.g., XRD, SEM, etc.) must be included under the experimental methods.

3.      The parameters (such as flow rate, distance between the needle and collector, and others) of electrospun procedure were not fully mentioned under the sup-topic ‘’synthesis of the electrospun dressings’’, its only voltage that was specified.

4.      The obtained results from TEM and XRD were not discussed convincingly.

5.      Some of the characterization techniques are missing such as FTIR, DLS, etc.

6.      Moreover, there are other several studies that are supposed to be performed on pristine and dual drug-loaded PVP nanofiber materials such as porosity analysis, water vapor transmission rate (WVTR), and mechanical characterizations.

Author Response

Reviewer 4

The research manuscript reported by Martín et al. demonstrated that the PVP-based nanofibers co-loaded with GO and ZnO nanoparticles are potential wound dressing materials for the treatment of infected wounds. Due to the lot of data that is missing, the article is not yet suitable for publication in pharmaceutics because there is a lot of data that is missing. However, the authors must address the following comments and suggestion before the paper can be considered for further review processes:

Response: we wish to thank the referee 4 for the positive comment to our work. A point-by-point answer can be found below:

Comment 1. The underlined subtitles under experimental methods should be numbered from 2.1. to 2.5. instead of underlined.

Response: we have now changed the format and included the numbering to these sections.

“2.1. Materials”

“2.2. Synthesis of GO/ZnO nanocomposites”

“2.3. Characterization of the nanomaterials”

“2.4. Synthesis of the electrospun dressings”

“2.5. SEM characterization of the electrospun discs”

“2.6. Antibacterial studies”

Comment 2. The reason for the application of each characterization technique (e.g., XRD, SEM, etc.) must be included under the experimental methods.

Response: we thank reviewer 4 for this comment, as it has helped to improve and clarify the experimental methods section as follows:

“2.3. Characterization of the nanomaterials

X-ray diffraction (XRD) analysis was performed to identify the crystalline phases presented by means of the diffraction patterns of the samples and nanocomposites by a Philips XPert 30XL with the CuK α radiation, ? = 1.5418 Å, under a voltage of 40 kV and a current of 40 mA. The diffraction data of samples were recorded for 2θ angles between 7 and 55. The transmission electron microscope used to study the surface morphology of the nanomaterials and to perform the Energy Dispersive X-Ray (EDX) analyses to confirm the presence of Zn, was a JEOL JEM 2010 coupled to a XEDS microanalysis system (Oxford Inca), with an accelerating voltage of 200 kV and a resolution between points of 0.25 nm. EDS (Energy Dispersive Spectrometer) analysis to determine the elemental compositions of the nanomaterials was performed using a Philips XL-30 conventional Scanning Electron Microscope coupled with an SDD-type EDS detector for microanalysis. TGA curves of the freeze-dried dispersions of the tested nanomaterials to study the thermal stability of the samples were acquired by using a TGA Q50 instrument (TA Instruments Company) from 30 to 900°C with a ramp of 10 °C/min under N2 or air using a flow rate of 90 mL/min and platinum pans.”

“2.5. SEM characterization of the electrospun discs

The fibers’ surface was analyzed by SEM (Philips XL-30 conventional Scanning Electron Microscope) operating at 10 kV. Fiber diameters were calculated by the ImageJ program.”

“2.6. Antibacterial studies

The well-diffusion method first proved the antibacterial ability of all the nanomaterials…”

Comment 3. The parameters (such as flow rate, distance between the needle and collector, and others) of electrospun procedure were not fully mentioned under the sup-topic ‘’synthesis of the electrospun dressings’’, its only voltage that was specified.

Response: we thank reviewer 4 again for this comment, as it has helped to complete the electrospinning parameters in the experimental methods section as follows:

“…electrospun mats. The experimental setup involved applying a voltage of 10 kV, maintaining a flow rate of 10 mL/min, using a needle with a diameter of 0.4 mm, and positioning the needle 80 mm away from the collector. After nine minutes…”

Comment 4. The obtained results from TEM and XRD were not discussed convincingly.

Response: we have reviewed the discussion section regarding TEM and XRD analyses and made several revisions to the text and Figure 1. However, we would greatly appreciate it if he/she could provide more specific details on what aspects he/she found lacking or problematic. Understanding his/her concerns in more detail would allow us to address them effectively and further refine the manuscript to even more closely meet the standards of rigor and precision expected in scientific research.

“TEM micrographs of GO/ZnO_1:1 and GO/ZnO_2:1 nanocomposites are shown in Figure 1a,b and Figure 1c,d, respectively. As it can be observed, ZnO nanoparticles form a star-shaped structure on the GO flakes. This specific morphology of the nanoparticles has been previously reported in the literature for ZnO composites at basic pH.28 Interestingly, the star-shaped structure seems to be more diffused in the GO/ZnO_2:1 nanocomposite compared to GO/ZnO_1:1, and it is also smaller in size. The higher GO:Zn(Ac)22H2O ratio could be influencing the formation of nanoparticles. Finally and most importantly, the ZnO nanoparticles were well distributed throughout the entire GO sheets in both GO/ZnO_1:1 and GO/ZnO_2:1 nanocomposites (Figure S1). This uniform distribution is crucial for achieving desired properties and functionalities in nanocomposite materials. Furthermore, EDX experiments confirmed the presence of Zn, when analyzing the ZnO nanoparticles (Figure S2).”

Figure 1. Representative TEM images of (a,b) GO/ZnO_1:1 and (c,d) GO/ZnO_2:1 nanocomposites. Scale bar (a,c): 0.5 µm. Scale bar (b,d): 100 nm. Red arrows indicate ZnO nanoparticles.

“The X-ray diffraction patterns from all the samples can be seen in Figure 2b. In the case of GO control sample, it showed a diffraction maximum close to 2θ≈10°, 9.88° (d-spacing: 8.8688Å) corresponding to the (002) plane,33–35 indicating the presence of oxygen-containing groups. In the case of the ZnO control sample, the characteristic diffraction maxima corresponding to the wurtzite type structure (JCPDS 891397)37 are observed36 at 31.73, 34.42, 36.46 that matched well with the (100), (002), (101) with hexagonal symmetry. The diffraction maxima of the graphene oxide and zinc oxide phases are noticed in the GO/ZnO composites. A variation of the relative intensities of diffraction maxima is observed due to the appearance of both phases, with a decrease of the intensities of the ZnO diffraction peaks as the proportion of GO in the nanocomposite is increased. The absence of any other peak confirms the purity of the nanocomposites. On the other hand, the absence of a diffraction peak (around 2θ≈26°), corresponding to the plane (002) of reduced graphene oxide (rGO) 38,39 indicated that the during synthesis process does not occur a reduction of GO, at least considerably or significantly to be detected by this technique.”

Comments 5 and 6. Some of the characterization techniques are missing such as FTIR, DLS, etc. Moreover, there are other several studies that are supposed to be performed on pristine and dual drug-loaded PVP nanofiber materials such as porosity analysis, water vapor transmission rate (WVTR), and mechanical characterizations.

Response: we would like to thank reviewer 4 for his/her valuable feedback regarding the characterization techniques used in our study. We appreciate his/her suggestion to include additional techniques such as FTIR, DLS, porosity analysis, water vapor transmission rate (WVTR), and mechanical characterizations.

While we acknowledge the importance of these techniques for an even deeper comprehensive material characterization, we would like to emphasize that our chosen characterization methods were selected strategically to align with the specific objectives and focus of our study. Our primary aim was to investigate the antibacterial properties of the electrospun PVP-based dressings containing GO/ZnO nanocomposites, as outlined in the title of our work.

Furthermore, while techniques such as FTIR and DLS could provide additional insights into the chemical composition and particle size distribution, respectively, our focus was primarily on confirming the presence and distribution of the nanocomposite components within the electrospun fibers. Moreover, that information is covered by TGA, EDS, EDX, XRD and TEM results.

Regarding porosity analysis, WVTR, and mechanical characterizations, we agree that these are important parameters to consider in wound dressing materials. However, again, since we focused on antibacterial properties, we prioritized studies in the presence of E. coli and B.subtilis accordingly.

This article presents a preliminary investigation of the synthesized dressings, providing a solid foundation that can be used for further results in the future. In this regard, we appreciate his/her suggestions and will consider them for future studies aimed at further comprehensively characterizing our electrospun PVP-based dressings.

Round 2

Reviewer 1 Report

Comments and Suggestions for Authors

no comment

Reviewer 4 Report

Comments and Suggestions for Authors

Dear,

All the comments and suggestions that were given are addressed accordingly. The manuscript can be accepted now for publication in pharmaceutics.